# On the Reliability of Surface Current Measurements by X-Band Marine Radar

**Katrin G. Hessner [1,\*], Saad El Naggar [2], Wilken-Jon von Appen [2] and Volker H. Strass [2]** 

[1] OceanWaveS GmbH, 21339 Lüneburg, Germany

[2] Alfred-Wegener-Institut Helmholtz-Zentrum für Polar- und Meeresforschung, 27515 Bremerhaven, Germany; Saad.El.Naggar@awi.de (S.E.N.); wilken-jon.von.appen@awi.de (W.-J.v.A.); Volker.Strass@awi.de (V.H.S.)

\* Correspondence: hessner@oceanwaves.de; Tel.: +49-4131-699-5822

**Abstract:** Real-time quality-controlled surface current data derived from X-Band marine radar (*MR*) measurements were evaluated to estimate their operational reliability. The presented data were acquired by the standard commercial off-the-shelf MR-based *sigma s6* WaMoS® II (WaMoS® II) deployed onboard the German Research vessel *Polarstern*. The measurement reliability is specified by an *IQ* value obtained by the WaMoS® II real-time quality control (*rtQC*). Data which pass the *rtQC* without objection are assumed to be reliable. For these data sets accuracy and correlation with corresponding vessel-mounted acoustic Doppler current profiler (ADCP) measurements are determined. To reduce potential misinterpretation due to short-term oceanic variability/turbulences, the evaluation of the WaMoS® II accuracy was carried out based on sliding means over 20 min of the reliable data only. The associated standard deviation $\sigma_{WaMoS} = 0.02$ m/s of the mean WaMoS® II measurements reflect a high precision of the measurement and the successful *rtQC* during different wave, current and weather conditions. The direct comparison of 7272 WaMoS® II/ADCP northward and eastward velocity data pairs yield a correlation of $r \geq 0.94$, with $|bias_\Delta| \leq 0.06$ m/s and $\sigma_S = 0.05$ m/s. This confirms that the MR-based surface current measurements are accurate and reliable.

**Keywords:** X-Band radar; marine radar current measurement; quality control; measurement reliability; accuracies; precision; WaMoS® II; vessel mounted acoustic Doppler current profiler

## 1. Introduction

Marine radars (MR) are designed for navigation and vessel traffic control. Depending on the physical environmental conditions given by precipitation, wind and waves, signatures of the sea surface commonly referred to as *sea clutter* become visible in the near range (<5 km) of the MR radar images. Regarded as a disruptive noise for navigational purposes, *sea clutter* is normally suppressed. Even though *sea clutter* signatures are well known, they are still not completely resolved, and are still under investigation both experimentally and theoretically. Nevertheless, it turns out that *sea clutter* includes valuable information on surface waves [1]. Following Bragg theory, *sea clutter* is caused by the backscatter of the transmitted electromagnetic waves from the short sea surface ripples in the range of half the electromagnetic wavelength (i.e., ~1.5 cm). Longer waves, such as wind sea (~10 m) and swell (~100 m), become visible as they modulate the *sea clutter* signal. Both surface currents and water depth affect the wave propagation [2,3]. As MRs image *sea clutter* simultaneously in time and space, this allows the derivation of multi-directional unambiguous wave information, surface currents, and (in shallow water) also water depth.

Driven by the growing need for precise information about waves and surface currents, commercially available MR-based wave and current monitoring devices, such as WaMoS® II, have

been developed [4–6]. Their capability and performance in a wide range of different applications, ranging from coastal applications [7–9] to vessel-mounted applications [10–12], have been proven.

The *sea clutter* observations of MRs typically range up to 3–5 km, with spatial and temporal resolutions on the order of 7.5 m and 2 s, respectively. This allows MRs to monitor waves longer than 15 m and current conditions over an area of several km$^2$ in real time. As *sea clutter* is caused by the *Bragg* backscatter of the transmitted electromagnetic waves from the short sea surface ripple waves (~2 cm), a minimum wind speed of 2–3 m/s is required for its presence [13]. In calm periods in the absence of ripples, no *sea clutter* can be observed, thereby preventing MR sea state and current observations. Also, signatures of rain or snow (*weather clutter*), or other features in the radar image not related to *sea clutter,* can disturb MR wave and current observations. These environmental limitations reduce the confidence and acceptance of the MR-based measurements, and therefore need to be treated carefully.

The aim of this paper is to assess the present status of the MR-based WaMoS® II system, focusing on its data usability with respect to reliability and accuracy. For this purpose, current measurements obtained onboard the German research vessel *Polarstern* during the Atlantic transit cruise PS113 [14] between Punta Arenas, Chile, and Bremerhaven, Germany, in May 2018 are used. The outline of the paper is as follows: In Section 2, we give a brief introduction on the methods used to estimate the accuracy and precision of fluctuating measurements. Section 3 describes the sensors used, with a focus on WaMoS® II. In Section 4, we present the WaMoS® II real-lime quality control (*rtQC*) used to specify data reliability. Observations made during the Atlantic transit cruise PS113 are presented in Section 5. Results of the accuracy estimation and comparison with acoustic Doppler current profiler (ACDP) measurements are presented in Section 6. Finally, in Section 7, we give a summary and draw conclusions.

## 2. Methods: Accuracy and Precision

A common method to evaluate the accuracy and precision of measurements is to perform a direct comparison of data sets from different sensors. In the case of MR-based current measurements, corresponding reference measurements from in situ sensors like ADCPs are used [15]. The underlying assumption of this approach is that both sensors observe the same property (*P*), and it is assumed that spatial and temporal homogeneity and deviations between the data sets can be related directly to inaccuracies in the measurements. However, this method of comparison is limited in that observed deviations do not automatically relate to inaccuracies of the measurement technique [16]. The biggest contribution to independent sensor deviations can be attributed to the different measurement locations of the sensors. For example, an ADCP delivers subsurface current measurements in a limited local volume, while MR-based observation represents current measurements at the sea surface over a spatial domain of several hundreds of square meters. Due to different current structures (e.g., wind-forced surface Ekman flow and geostrophic current shear extending deeply over a large part of the water column), vertical homogeneity is not given at all times. Ref. [17] found that 80%, or more, of the observed deviations between ADCP and HF radar current measurements on the West Florida shelf were associated with horizontal and vertical separation between the measurements.

In addition, the informative value of the direct comparison of two independent data sets might be misleading, as it completely neglects the natural variability of the current as a vector, consisting of mean, oscillatory and chaotic contributions. This makes the results more difficult to compare and properly interpret [18]. Therefore, we use a combination of methods to evaluate the quality, reliability, precision, and accuracy of MR surface current measurements.

For practical handling of a fluctuating quantity, *P*, its temporal averages $\overline{P}$ over a suitable period (averaging time $\tau$) are used. This allows to describe *P* as $P = \overline{P} + P'$, where $P'$ is the fluctuation with $\overline{P'} = 0$. Based on this assumption, the resulting measurement is represented by the average $\overline{P}$, which depends, among other things, also on the used sampling and averaging intervals.

In this paper, we aim to evaluate the general performance of WaMoS® II data by directly comparing both the mean current, $\overline{U}$, and the corresponding standard deviation, $\sigma_U$, representing the

short-term oceanic fluctuating component of the current. To evaluate the accuracy of the WaMoS® II measurements, a direct comparison of $\overline{U}$, with the corresponding ADCP measurements is carried out, where the accuracy is described by correlation coefficient ($r$), bias ($\overline{\Delta}$) and standard deviation ($\sigma_\Delta$) of the difference:

$$r = \frac{\sum_{i=1}^{N}\left(X_i - \overline{X}\right)\left(Y_i - \overline{Y}\right)}{\sqrt{\sum_{i=1}^{N}\left(X_i - \overline{X}\right) \cdot \sum_{i=1}^{N}\left(Y_i - \overline{Y}\right)}} \tag{1}$$

$$\overline{\Delta} = \frac{1}{N}\sum_{i=1}^{N}\Delta_i, \text{ with } \Delta_i = X_i - Y_i \tag{2}$$

$$\sigma_\Delta = \sqrt{\frac{1}{N-1}\left(\sum_{i=1}^{N}(\Delta_i - \Delta)^2\right)} \tag{3}$$

where $\overline{X} = \frac{1}{N}\sum_{i=1}^{N}X_i$ and $\overline{Y} = \frac{1}{N}\sum_{i=1}^{N}Y_i$ represent the mean measurement of the data sets $X$ and $Y$, respectively. The resulting combined standard deviation is defined as $\sigma_\Delta = \sqrt{\sigma_X^2 + \sigma_Y^2}$. Assuming that the measurement errors of the two sensors are uncorrelated and of equal magnitude, the individual (single) standard deviation $\sigma_s = \sigma_X = \sigma_Y$, and can hence be estimated by:

$$\sigma_S = \frac{1}{2}\sqrt{2}\,\sigma_\Delta. \tag{4}$$

Note that $r$, $\overline{\Delta}$, and $\sigma_S$ include deviations related to horizontal ($\sigma_{\Delta h}$) and vertical ($\sigma_{\Delta v}$) gradients, as well as temporal variation ($\sigma_t$) of $P$, which are not related to inaccuracies of the measurement device. Using the mean instead of the instantaneous measurements leads to statistically more stable and reliable results as the effect of uncorrelated natural variability is minimized.

To evaluate the precision of an individual sensor itself, we use a more general approach. This approach is based on statistical analysis of a property $P$, represented by a statistical population $\{P_1, P_2, \dots, P_N\}$. The precision of the measurement of $P$ can be estimated by the standard deviation $\sigma(\overline{P})$ of the mean $\overline{P}$, which is given by:

$$\sigma(\overline{P}) = \sqrt{\frac{1}{N-1}\sum_{i=1}^{N}\left(P_i - \overline{P}\right)^2} \tag{5}$$

where $i = 1, N$ denotes individual values over the averaging interval $\tau$.

Following this strategy, the precision of a measurement is estimated by the standard deviation $\sigma_P$ of the mean $\overline{P}$. Using an averaging interval of $\tau = 20$–30 min in combination with typical update rates of WaMoS® II measurements ranging between 1–3 min allows us to obtain a sufficient number of independent measurements, and hence gives statistically significant results for our investigation.

## 3. Data

The data used for the WaMoS® II-ADCP comparison were acquired on board *Polarstern* during the Atlantic transit cruise PS 113 (May 2018) [14].

### 3.1. Sigma S6 WaMoS® II

The MR-based measurements were carried out by the *sigma* S6 WaMoS® II system. This standard commercial, off-the-shelf system consists of a high-speed video digitizing and storage device, which can be interfaced to most conventional analog and digital navigational X-Band radars. The *sigma* S6 system technology can be supplied with different software packages for various real-time applications like small target detection, oil spill detection and ice navigation and monitoring, as well as real-time sea state and current measurements (WaMoS® II).

The WaMoS® II system can be operated from fixed platforms and coastal sites, as well as from moving vessels. For the latter application, the horizontal vessel motion needs to be compensated. The large vertical beam width of MRs, the range of which is normally between 20° and 25°, depending on the used radar type, ensures the ability to scan the sea surface even when the ship is pitching and rolling [19]. Hence, we assume that vessel motions like pitch, roll and heave have no critical influence on the WaMoS measurements.

The horizontal vessel movement can be removed during data processing, either in the space-time or in the wave number-frequency domain. The compensation in the wave number-frequency domain requires that the vessel movement over ground is constant (no variation in speed or course) during radar data acquisition given by number of individual radar images times radar repetition rate. In this case, the vessel movement is related to a fixed Doppler shift ($\vec{k}\,\vec{V}_{ship}$), and can be separated from the Doppler shift related to the surface current ($\vec{k}\,\vec{U}_{current}$), where $\vec{k}$ is the wave number vector. The motion compensation in the space-time domain is performed by georeferencing [15,20]. Using GPS ship position and heading (gyro), orientation and position are estimated for every radar pulse. When transforming the sea clutter information from polar coordinates to Cartesian image sequences, each point of the resulting analysis area corresponds to a fixed position relative to the earth, independent of how the vessel is moving during the acquisition time. This method requires more computing time and limits the area available for analysis, but is independent of the ship movement. However, in cases when the vessel is moving very fast (>20 kn), this method might fail, and this occurs when the analysis area moves out of the radar view field. In both cases, very precise vessel heading is required as the error due to misalignment is amplified by the vessel speed [10]. For this application, WaMoS® II processing was set to *georeferencing* mode, as the vessel speed of *Polarstern* was <12 kn, in general. With the sampling strategy of 64 images per individual WaMoS® II measurement and a radar rotation time of 2.5 s, the maximum expected offset during data acquisition is about 1 km, and this is acceptably small compared to the WaMoS® II radar view range of 3 km.

The WaMoS® II system onboard *Polarstern* is connected to an analog SAM Radarpilot 1100 (9.4 GHz), with a rotation rate of $RPT$ = 2.5 s. This radar is dedicated to the WaMoS® II application (in the following, this is referred to as the WaMoS® II radar). The mounted 5 ft antenna provides 1.5° angular resolution. Running in short pulse mode, with a pulse length of 80 nsec, the transceiver delivers data with 12 m range resolution. By oversampling of the radar raw data in direction and range, the *sigma* S6 digitizer delivers radar information with an angular resolution of 0.35° and 5.62 m in range (26.7 MHz). The WaMoS® II radar view field covers a range for 0.303–2.356 km from the antenna, with the second quadrant sector blanked due to the mast construction (Figure 1).

For one individual WaMoS® II measurement on board *Polarstern*, 64 consecutive radar images were analyzed, so that the WaMoS® II results represent temporal means of 2.67 min (64 × 2.5 s). To overcome the effects of the directional dependency of the wave imaging in radar images from radar look direction relative to wave and wind direction [10], the WaMoS® II analysis areas were placed all around the vessel. Figure 1 shows a vessel-oriented radar image, which was obtained by *sigma* S6 WaMoS® II on 12 May 2018, 12:00 UTC. At that time *Polarstern* was sailing northeastwards (42°) at 12 kn (6.2 m/s), while the wind was blowing from 271°, at about 14.4 m/s. The color refers to the measured radar return: black meaning no return level, and white indicating the maximum level. The radar return is digitized to 12 bits, which allows a signal strength ranging from 0–4095. To ensure no information is lost due to clipping at the lower limits, the digitizer is set below the noise level of the system. For the *Polarstern* system, a mean noise level of 500 was determined during system installation. To avoid reflections from the vessel superstructure in the near range, the system starts sampling after a dead range of 300 m. The analysis areas are the three grey rectangles indicating size (128×256 pixels) and alignment (35°, 255°, 325° relative to vessel heading) of the *sigma* S6 WaMoS® II (Figure 1). To overcome the directional dependency of the wave imaging in the radar images, the individual wave spectra of each analysis area are averaged. From the resulting spatially averaged spectrum, statistical wave parameters such as significant wave height ($H_s$), peak wave period ($T_p$), peak wave direction

($\theta_p$), etc., are derived. The WaMoS® II update rate onboard *Polarstern* is approximately 3 min, given by the time taken for 64 images to be acquired, multiplied by the radar repetition rate of 2.5 s.

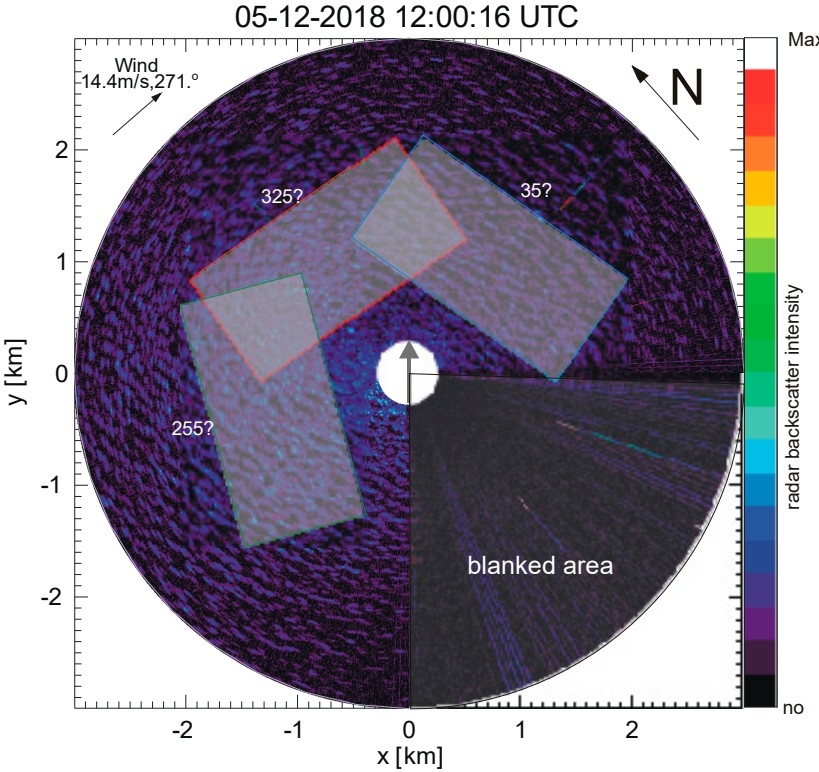

**Figure 1.** Vessel head-oriented WaMoS® II radar image acquired onboard the *Polarstern* on 12 May 2018, 12:00 UTC. The color scale refers to strength of the received radar return. The highlighted boxes indicate size and alignment of the three WaMoS® II wave analysis areas relative to the radar view field. The arrow in the center indicates the orientation of the *Polarstern*, which was moving at 12 kn and a course of 43° during data acquisition, and north is indicated by the arrow in the top right.

From the measurements of the phase speed (*c*) of the captured surface waves, the underlying ocean surface currents ($\vec{U}_s$) are derived by identifying deviations from the known dispersion relations of surface waves. Assuming that $\vec{U}_s$ is small compared to *c*, the depth-weighted effective surface current is given by:

$$\vec{U}_s(k) = 2k \int_0^H \vec{U}(z) \exp(-2kz)dz \qquad (6)$$

where $\vec{U}(z)$ is the vertical current profile, with z being positive downwards and *H* being the water depth. As the influence decays exponentially with depth, the resulting current $\vec{U}_s$ represents a vertical average of the ocean currents within the wave-influenced surface layer $D_W$ [21]. $D_W$ varies depending on the wavelength ($\lambda = 2\pi/k$) and height of the captured waves. On average, $D_W$ is assumed to range between 3 and 10 m depending on the predominant wave length, and this will be shallower for short wind sea waves than for long swell waves [15].

### 3.2. ADCP

As a reference, data from a vessel-mounted acoustic Doppler current profiler (ADCP) type Ocean Surveyor from Teledyne RD instruments [22] were used. Its transducers/receivers, operating at a nominal frequency of 150 kHz, are mounted in the hull of *Polarstern*, about 11 m below the water line. It was working in long-range mode with a vertical cell size of 4 m, a blanking distance of

4 m, and a maximum range of ~320 m. Heading, pitch, and roll from the ship's inertial navigation system (GPS and magnetically constrained "gyro") were used to convert the ADCP velocities to earth coordinates. The accuracy of the ADCP velocities mainly depends on the quality of the position fixes and the ship's heading data. Further errors stem from a misalignment of the transducer with the ship's centerline. The ADCP data were processed using the Ocean Surveyor Sputum Interpreter (OSSI) developed by GEOMAR, Helmholtz Centre for Ocean Research, Kiel ([22]), which corrects for a possible misalignment between the ADCP transducer orientation and the ship's forward direction.

To avoid interference with vessel-induced currents, the ADCP measurements are averaged over the 20–50 m depth range. For the data comparison, quality controlled ADCP current data with averages over 2 min were used. The quality filter is based on the statistical analysis, where data outliers exceeding the range of mean value and standard deviation of surface velocity are neglected. The ideal-theoretical precision $\sigma_{ADCP(ideal)}$ of the ADCP measurements can be estimated from the single ping/bin standard deviation of $\sigma_{ADCP(SP)} = 0.3$ m/s, given by the ADCP manufacturer. Neglecting natural variability and assuming vertical and temporal homogeneity and independence over 20–50 m (7 depth bins) and 2 min (100 pings), results in $\sigma_{ADCP(ideal)} = \sigma_{ADCP(SP)}/\sqrt{N} = 0.0113$ m/s.

## 4. WaMoS® II Quality Control

The detection of surface waves in radar images is sensitive to data acquisition and environmental conditions. To create *sea clutter*, a minimum wind speed is required [13]. Furthermore, a minimum wave height is required to significantly modulate the *sea clutter* information. In case of insufficient wave signatures in the radar images, the system cannot deliver reliable information. To indicate the reliability of a *sigma* S6 WaMoS® II measurement, real-time quality control (*rtQC*) is implemented. During processing, the *rtQC* is carried out in different steps, which are also summarized in Figure 2:

- Data acquisition check: This check verifies if all mandatory information is available.
- *Sea clutter* checks: In this step, the radar raw input is controlled. In cases of insufficient *sea clutter* information due to no sea state, rain, very low wind conditions (<3 m/s) or missing data sections, the data set is marked with a quality identifier *IQ* of $0 < IQ < 9$.
- SNR-checks: Here, the quality of the separation of signal and noise within the image spectrum is used to evaluate the quality of the current estimate and wave filtering. Data sets which do not pass these checks have potentially unreliable current estimates and are marked with $10 < IQ < 400$.
- Wave system checks: When deriving the standard sea state parameters from the frequency direction spectrum $E(f, \theta)$, the number of individual wave systems is determined. In cases of more than 5 wave systems, $E(f, \theta)$ can be regarded as too noisy to give reliable results. In these cases, the resulting data sets are marked with $400 < IQ < 1000$.
- Data basis check: For spatial and temporal averages, a minimum of individual measurements is required to obtain a statistically stable result and hence trustful data. All mean data sets with less measurements are marked with $0 < IQ < 10$.

The results of the different *rtQC* tests are accumulated and summarized in an individual *IQi* value, which is assigned to the measurement areas (i) (Figure 2). The *IQ* values are binary numbers where each digit (0 or 1) corresponds to whether the individual measurement has passed or failed a particular test. These binary numbers are then converted to decimals, which is what is given as output and discussed here. Based on the individual *IQ* value, it is possible to separate reliable data which pass the tests from unreliable data which do not pass all of the tests. Furthermore, the binary structure of the *IQ* makes it possible to backtrack the individual results of the different tests, and therefore enables validation of the significance of an individual test and the adjustment/optimization, if necessary, of thresholds for particular installations. For example, if the sea clutter test yields parts of the analysis area that contain rain signatures (see Figure 1), this leads to *IQ* = 4. If in the SNR check the data does not pass the significance test so that *IQ* = 20, this is added to give *IQ* = 24. Furthermore, if the wave system check failed as the resulting spectrum is too noisy to identify significant individual wave systems,

$IQ = 100$ is added. The accumulated result is then $IQ = 124$, indicating unreliable measurements due to insufficient radar data because of rain. For the presented data set, it turned out that the sea clutter checks do not indicate unreliable data on their own. This means that partly missing or disturbed sea clutter information alone does not automatically lead to identification as unreliable data. Only in combination with the subsequent quality checks can the results be regarded as unreliable. Even when the results of the sea clutter checks do not explicitly indicate insufficient data, it reveals the potential cause of measurement failures. This will be discussed in more detail with respect to rain signatures later on. For practical use a decimal $IQ$ valid limit of 10 is set.

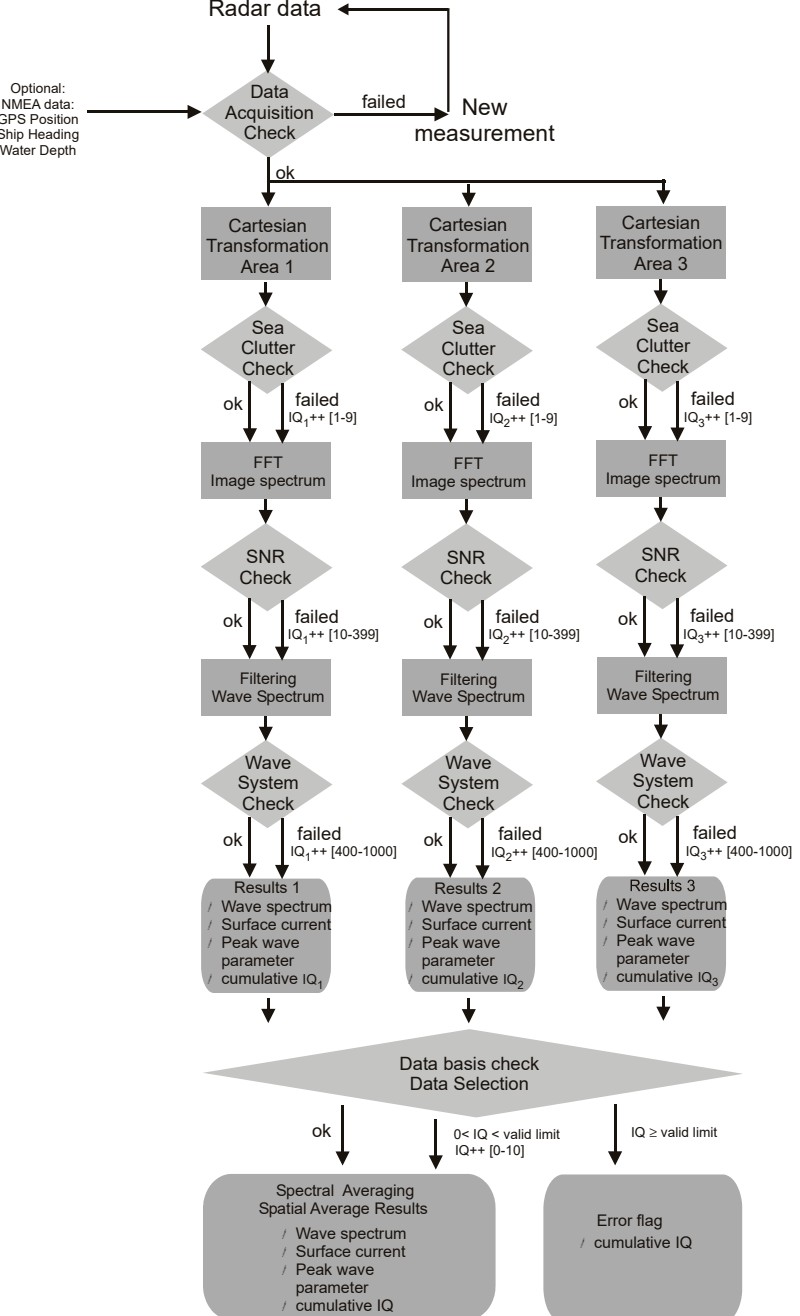

**Figure 2.** Flow chart of the WaMoS II real-time quality control (rtQC) The different quality checks (diamonds) are carried out with in the levels of the WaMoS processing chain. Boxes indicate key steps in the processing from the radar images in polar coordinates to the resulting wave spectra, peak wave and current parameter.

Figure 3a shows a radar image acquired during a calm period with wind speed <3 m/s. Due to the absence of ripple waves (~3 cm) no *sea clutter* is visible. Figure 3b shows a radar image acquired during heavy rain. The potential *sea clutter* is covered by the rain signatures visible as patches of high backscatter intensity generated by the rain drops in the air. In both cases, no sufficient *sea clutter* is visible, and the corresponding *rtQC* results in *IQ* > 400.

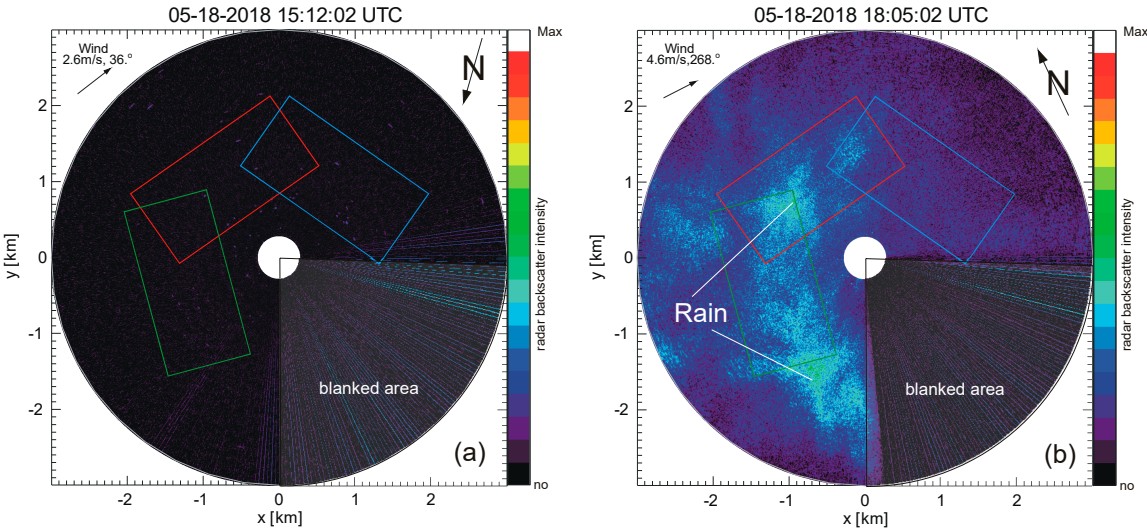

**Figure 3.** Radar images showing insufficient *sea clutter* information for wave and current retrieval. (**a**) Calm wind condition showing no *sea clutter*. (**b**) Heavy rain condition, where rain signatures covers *sea clutter* information. The color code refers to strength of the received radar return. The highlighted boxes indicate size and alignment of the three WaMoS® II analysis areas.

Figure 4 shows radar images under limited conditions due to moderate rain. The rain signatures are spatially localized (Figure 4a) or are weak (Figure 4b) and allow the detection of wave signatures and therefore also the wave and current analysis. The evaluation of the resulting data suggested that some of the wave parameters are more robust than others with respect to limited *sea clutter* information. The analysis shows that the directly measured parameters like wave period ($T_p$), wave direction ($\theta_p$), and surface current ($\vec{U}$) are more robust than the indirect measurement of significant wave height ($H_s$). As long as no other *rtQC* check has been failed, these direct wave parameters can be assumed to be reliable when *IQ* < 10. This does not apply to the indirect estimates of $H_s$, which are based on the accurate determination of the signal to noise ratio (SNR). Missing *sea clutter* information leads immediately to missing signal intensity and hence to a lower SNR, which results in a decrease in $H_s$.

It needs to be stressed that the *rtQC* is performed in real time during data processing, and does not require post processing. Furthermore, the identification of insufficient radar information is carried out independently of external information on rain and wind. In cases where the internal WaMoS® II *rtQC* identifies insufficient *sea clutter* (*IQ* > 10), this information is aligned with wind information (if available) to give the combined information, to indicate that currently MR measurements are not possible due to the lack of sufficient wind. This keeps the WaMoS® II *rtQC* independent of additional external information and their reliability. The relation between *IQ* and wind information is an additional piece of information for the user and makes it possible to evaluate the actual threshold of the wind speed for the particular WaMoS® II set up (used radar, installation geometry, etc.).

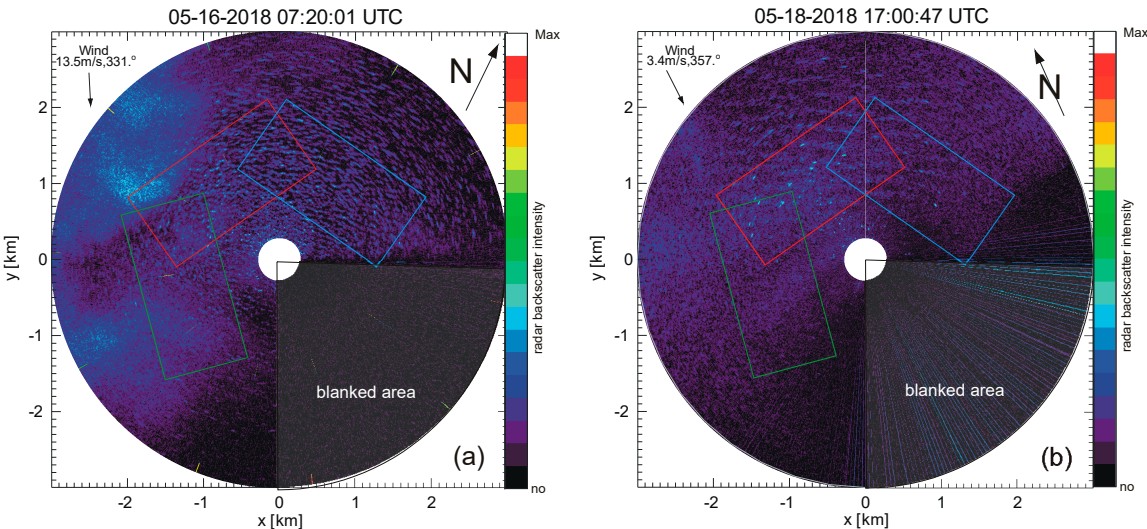

**Figure 4.** Radar images showing limited sea clutter information for wave and current retrieval. (**a**) Rain signature partly obscures *sea clutter*. (**b**) Weak rain signatures blur the *sea clutter*.

## 5. Observations During the Cruise

Figure 5 shows the cruise track of *Polarstern* during *PS113* for May, 2018 outside of exclusive economic zones (*EEZ*, 200 miles). The WaMoS® II recording started when *Polarstern* left the *EEZ* of Argentina, 11 May 2018. During the cruise different current regimes from wind-forced to density-driven currents up to more than 2 m/s were encountered. Besides different current regimes, various environmental conditions were experienced. These ranged from a storm event at the beginning of the cruise (May 12/13), with wind speeds up to 20–25 m/s and sea states up to 6–7 m significant wave height ($H_s$), to calm (wind speed <3 m/s) and rainy periods. The latter give the possibility of validating the proper *rtQC*. Even though no reference data for the WaMoS® II sea state measurements were available, unreliable data can be identified in the data set, as during insufficient *sea clutter* conditions, the peak wave direction and current shows an unrealistically high variance.

During the cruise, it turned out that the other X-Band radar onboard *Polarstern*, which is used for navigation, interfered strongly with the WaMoS® II radar. This led to partly corrupted radar image acquisition and gaps in the time series. These corrupted radar data were identified by WaMoS® II *rtQC*. To minimize and evaluate the impact of the radar interferences on the WaMoS® II measurements, corrupted images or sectors were replaced by blanked data, and such data sets are marked with *IQ* = [1 or 2], depending on the amount of missing data in the analysis sequence. This allowed us to evaluate the impact of missing sectors on the general performance of the system. It turned out that, when sufficient *sea clutter* was visible in the other parts of the image and no other *rtQC* test failed (*IQ* > 10), the direct measurement was undisturbed and reliable. The missing signal leads to an underestimation of the indirect measurement of $H_s$, which is indicated by *IQ* > 0.

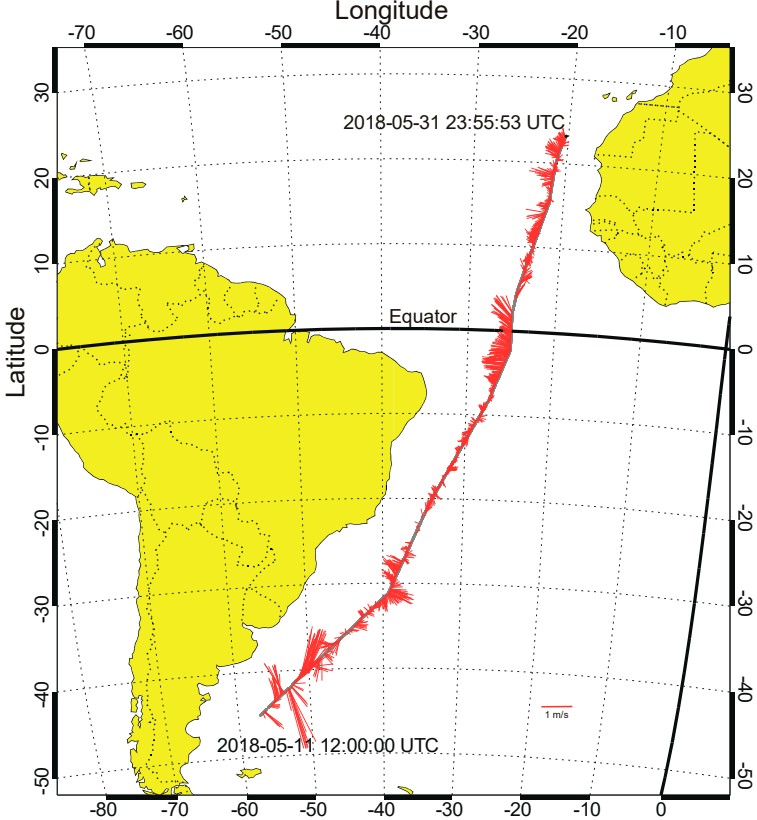

**Figure 5.** Cruise track (grey) of *Polarstern*. Stick plot (red) indicating the surface currents observed by WaMoS® II during PS113 (May data only). The length of the sticks is related to current speed, orientation represents current direction (going to).

The resulting data (Figure 6) analysis proved the performance of the *rtQC*. It successfully identified data sets with insufficient *sea clutter* during rain or no sufficient wind (>3 m/s) conditions, which are marked grey. Cases with insufficient wind speed were confirmed by independent direct wind speed measurements (German Weather Service, DWD) onboard. Rain cases were confirmed by visual observations and visual inspections of the corresponding radar image. Figure 6 shows the time series of wind speed (top panel in turquoise) and WaMoS® II current speed and direction (middle and lower panel). WaMoS® II data with *IQ* < 10 are marked in red, while data with *IQ* > 10 are marked in grey. The data sets, which were acquired during very low wind speeds (<3 m/s), show unrealistic scatter in the current speed and direction. These data sets were successfully identified (*IQ* > 10) by the WaMoS® II *rtQC*, and hence are displayed in grey. This evaluation confirms that a minimum wind speed of 3 m/s is required for reliable WaMoS® II measurements. In cases of too low wind speed, the radar images contain only random noise rather than wave information. Without the stringent quality control, the WaMoS algorithm outputs current solutions based on random noise which are completely unrelated to the real current conditions (e.g., May 18th). Please note that the WaMoS® II *rtQC* is independent of the wind measurements; hence, it is independent from their availability, accuracy and reliability.

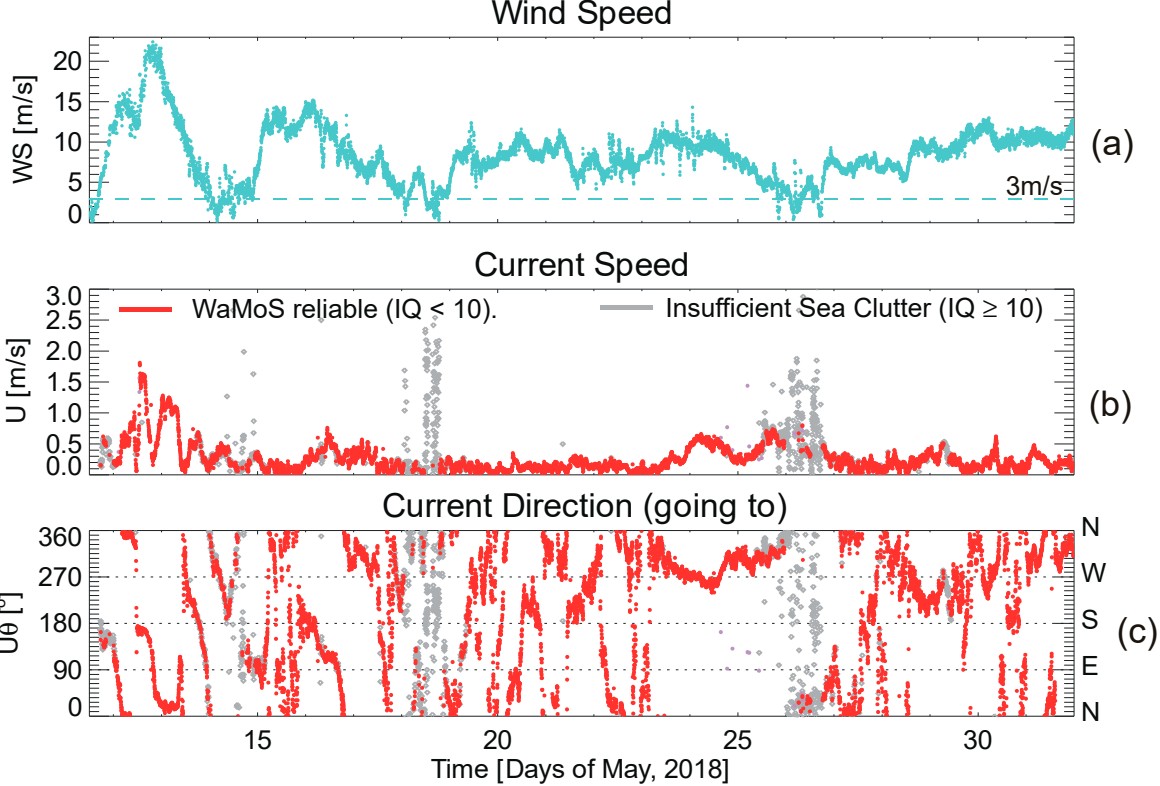

**Figure 6.** Time series of wind speed (**a**) and WaMoS® II surface current speed (**b**) and direction (**c**). Data which passed the WaMoS® II *rtQC* with *IQ* < 10 are in red, data with *IQ* ≥ 10 are assumed to be unreliable and are shown in grey.

For the May 2018 period shown, WaMoS® II carried out 9727 individual instantaneous measurements. From this data set, about 80% pass the *rtQC* with *IQ* < 10 and can be accepted as reliable for direct wave and surface current measurements. The rest of the data is identified as unreliable because of insufficient *sea clutter* due to interferences with the navigation radar and/or environmental conditions (no sufficient wind, rain, very low sea state). About 10% of the WaMoS® II data had reduced quality, with 10 < *IQ*< 400, characterized by noisy wave spectra. These data sets may include valuable information, but no statistically reliable estimates of integrated wave parameters can be derived. Especially rain induced noise interferes with *Hs* estimates, while more robust direct measurements like $T_p$ or current $\vec{U}$ might contain valuable information. The final 10% of the data set with *IQ* > 400 does not contain any sufficient radar signals for WaMoS® II processing.

The results of the WaMoS® II surface current measurements were compared with the ADCP measurements (Figure 7). The visual comparisons of the measurements demonstrate the general agreement of the WaMoS® II surface and ADCP subsurface measurements, with exception of the equatorial region, where a significant vertical current shear does not allow a direct comparison.

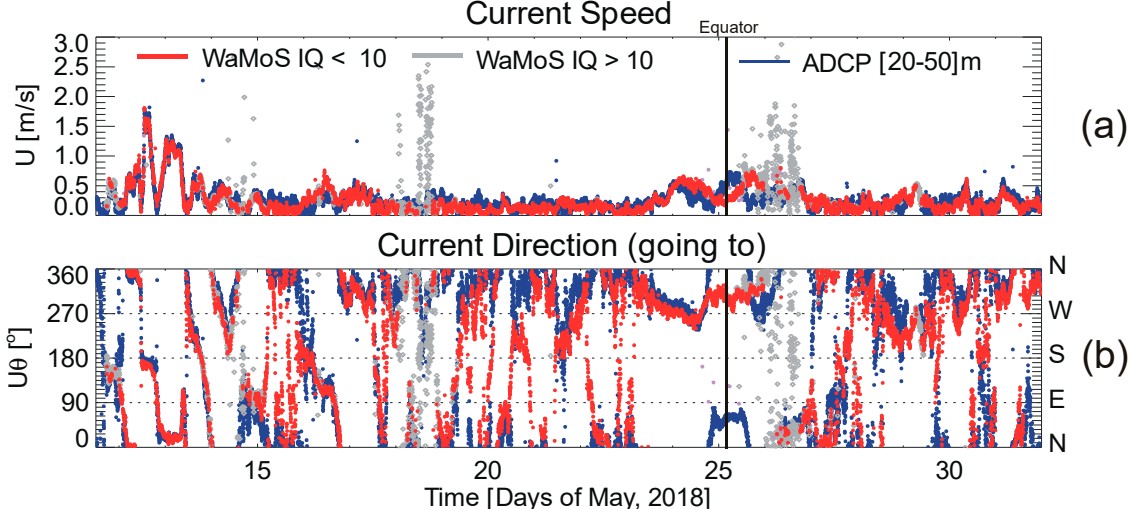

**Figure 7.** Time series of the current speed (**a**) and current direction (**b**) as derived by WaMoS® II (red/grey) and ADCP (blue) on board *Polarstern* during PS113 cruise. The grey values indicate WaMoS® II marked as unreliable (*IQ* ≥ 10). The vertical line at May 25th marks the time when the equator was crossed.

The best agreements in current speeds and directions between WaMoS® II and ADCP were observed at the beginning of the cruise (May 12th–14th) in mid-latitudes (~40° S). Oceanographically, this region is characterized by the opposing northward oriented *Falkland-Malvinas Current* and the southward flowing *Brazil Current*. The area where both currents meet, the *Brazil Falkland-Malvinas Confluence*, is recognized as one of the most energetic in the world's ocean, with large-amplitude meanders and mesoscale eddies [23]. WaMoS® II as well as the ADCP, observed almost identical currents with maximum speeds up to 2 m/s, strongly varying in speed and direction. Leaving this zone, a region with current speed <0.5 m/s was passed. Here small deviations between surface WaMoS® II and sub-surface ADCP measurements can be observed. This is most likely due to vertically inhomogeneous current conditions. In the region of the equator (±2°, ~May 25th), the surface current measured by WaMoS® II and the subsurface current recorded by ADCP deviate. This deviation is primarily caused by the Equatorial Under Current (EUC) that occupies the depth range of 30–250 m with its strong eastward-directed velocities [14], opposing the north-westward-directed wind drift of the surface layer.

## 6. Results

In this section, we present the comparison of the quality-controlled WaMoS® II surface current data (*IQ* < 10) with the quality-controlled ADCP subsurface current data. The agreement of both data sets is estimated from the following statistical parameters: Bias, correlation coefficient (*r*) and standard deviation ($\sigma_\Delta$, $\sigma_s$). For both data sets (WaMoS® II and ADCP), standard deviation of the mean ($\sigma_{WaMoS}$, $\sigma_{ADCP}$) for the individual measurements was determined over an averaging interval of 20 min. For the comparison, the data obtained near the equator was excluded, because vertical homogeneity was not given there [14]. Finally, 7272 individual data sets pass the *rtQC* and are used for this evaluation.

Figure 8 shows the direct comparison of the quality controlled WaMoS® II surface and ADCP subsurface current measurements for the eastward (*UE*) and northward (*UN*) components. For both components (*UE* and *UN*), the statistical results (Figure 8a,c) are in the same range (*r*: 0.94, 0.97; bias = −0.02 m/s, −0.06 m/s and $\sigma_s$: 0.05 m/s, 0.05 m/s, respectively). For the *UN* component, the correlation *r* as well as $bias_\Delta$ and $\sigma_s$ are slightly higher. This is most likely related to the fact that higher absolute values of *UN* with speeds up to 1.5 m/s were observed, while *UE* speeds remained below 0.7 m/s during the entire cruise. The corresponding histograms of the signed differences ($\Delta UE$

and $\Delta UN$) between the WaMoS and ADCP current components are shown in Figures 8b and 8d. Both distributions can be fairly well approximated by a Gaussian distribution ($ae^{-(x-b)^2/c^2}$, with $a$ the height, $b$ the center, $c$ the width (the standard deviation) of the Gaussian distribution). Given the fact that WaMoS and ADCP measure in different depths, and that ocean currents tend to be vertically sheared and to veer with depth geostrophically, following the Ekman spiral of the wind driven flow or the Stokes drift associated with surface waves, some differences between the two measurements are expected. Hence, a bias of the found magnitude and the slight deviations of the histograms from Gaussian shape do not necessarily signify a measurement error.

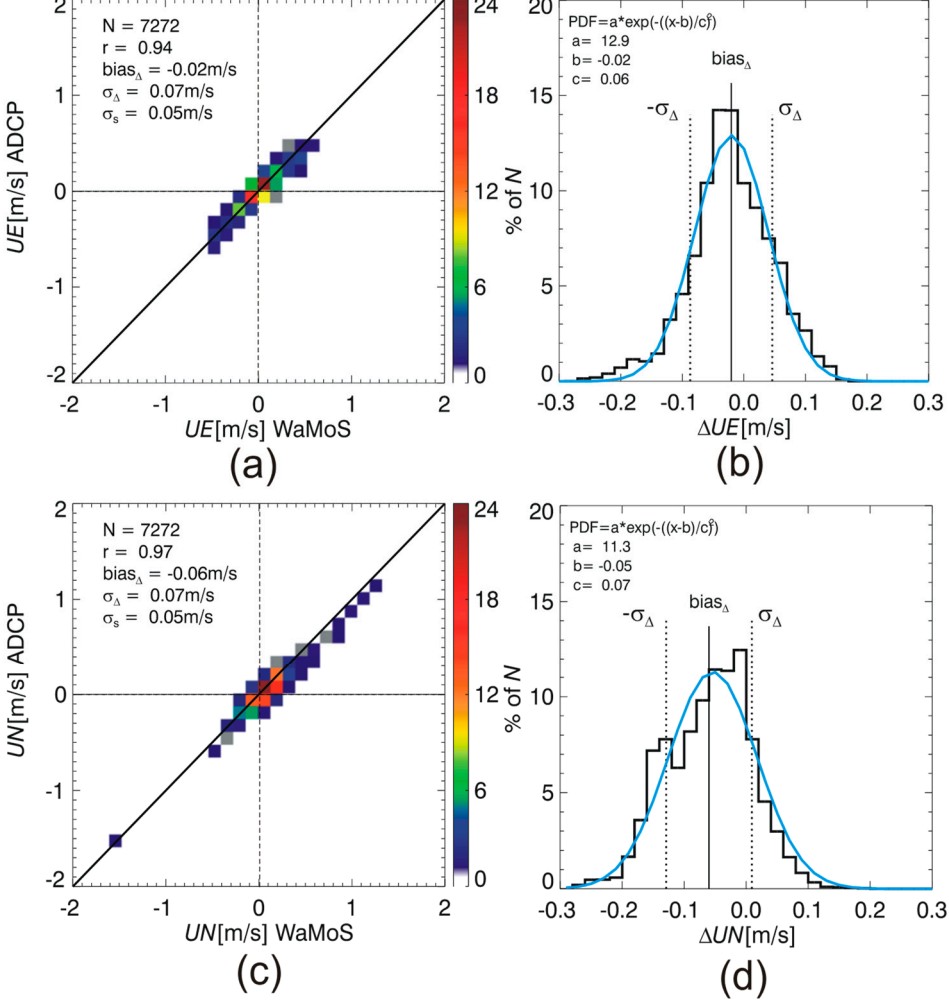

**Figure 8.** Comparison of the Eastward (*UE*) and Northward current components (*UN*) of WaMoS® II surface and ADCP subsurface (mean of the 20–50 m depth range) currents. (**a,c**) Scatter plot of WaMoS versus ADCP. (**b,d**) Histograms (black) and PDF (blue) of the differences between WaMoS and ADCP. *N* gives the number of data pairs, *r* the correlation coefficient, $\sigma_s = \frac{1}{2}\sqrt{2}\,\sigma_\Delta$ the standard deviation. The bin width is 0.1 m/s in the scatter plots and 0.02 m/s in the histograms.

The comparison of the absolute current speed ($US = \sqrt{UE^2 + UN^2}$) (Figure 9a,b), reveals the same agreement (*r*: 0.96 and $\sigma_s$: 0.05 m/s) with an almost perfect Gaussian distribution and no significant bias (−0.0004 m/s) of the differences. For the current direction (*Uθ*, Figure 9c,d) the correlation is slightly lower (*r*: 0.87). The bias of −6.88° suggests that the disagreement between the two systems is mostly due to differences in the orientation of the determined currents. However, we cannot completely rule out at present that one of the two measurement systems or both are subject to some small systematic errors. A bias in the same range as in our observations is reported from most

studies comparing MR-derived surface current estimates with estimates from other devices (e.g., [10]). It is also possible that effects from a possible misalignment of the ADCP transducer are not completely removed during the ADCP data processing.

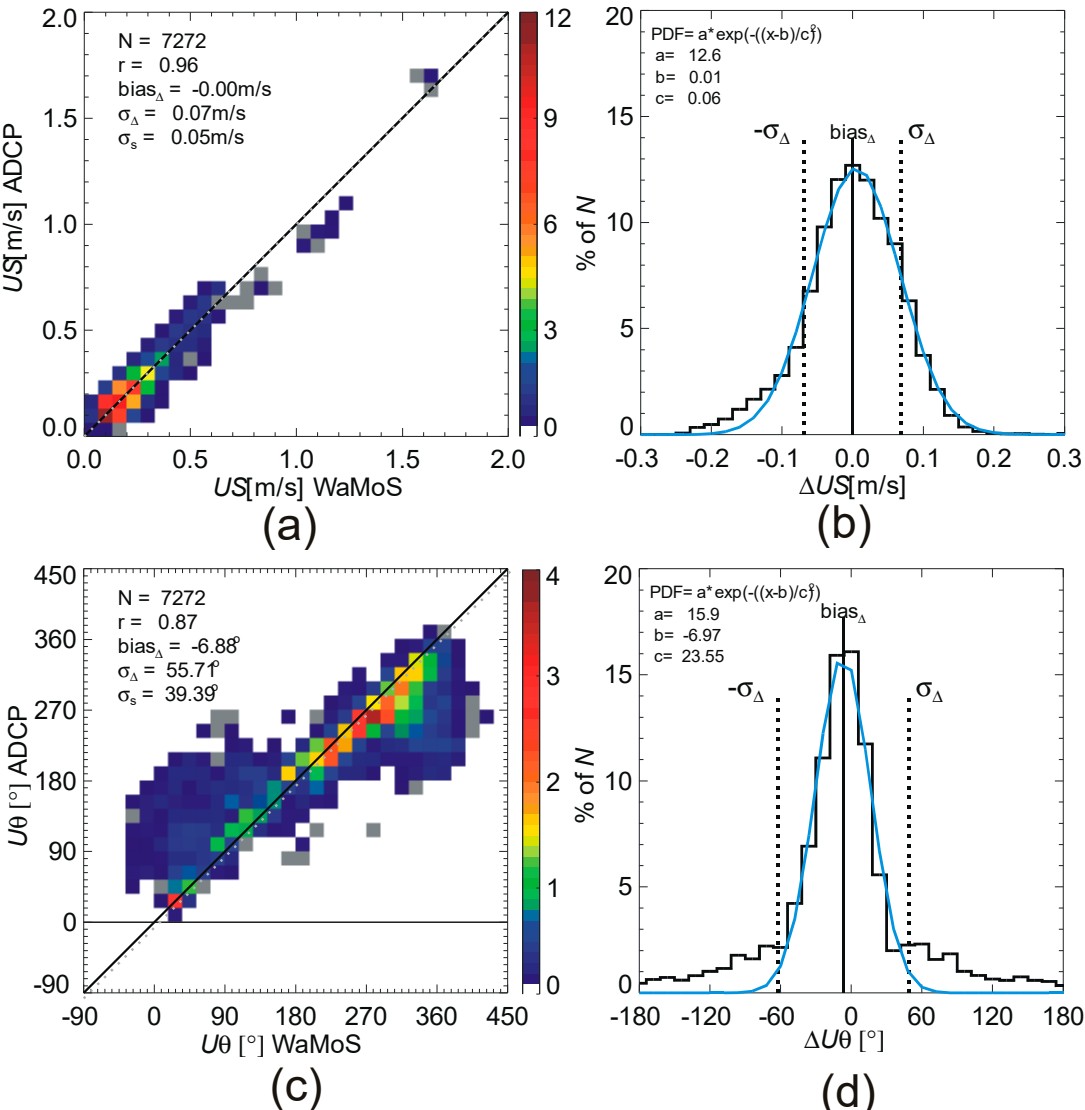

**Figure 9.** Comparison of the current speed (*US*) and current direction (*U*θ) of WaMoS® II surface and ADCP subsurface (mean of the 20–50 m depth range) currents. *U*θ = 0 refers to northward current. Please note that due to the angle discontinuity at North (360°/0°), *U*θ covers the range of [−90:450°]. (**a**,**c**): Scatter plot of WaMoS versus ADCP. (**b**,**d**): Histograms (black) and PDF (blue) of the differences between WaMoS and ADCP. *N* gives the number of data pairs, *r* the correlation coefficient, $\sigma_s = \frac{1}{2}\sqrt{2}\,\sigma_\Delta$ the standard deviation. The bin width for *US* is 0.1 m/s in the scatter plot (**a**) and 0.02 m/s in the histogram (**b**). The bin width for *U*θ is 16° in the scatter plot (**c**) and 12° m/s in the histogram (**d**).

Histograms of the absolute deviations $\left|\Delta(UE)\right|$ and $\left|\Delta(UN)\right|$ (black) and of the individual standard deviations for WaMoS (red) and ADCP (blue) for both current components are shown in Figure 10. Again, the measurement differences for both current components reveal approximately the same distribution, covering the range up to 0.2 m/s and standard deviation of $\sigma_\Delta = 0.07$ m/s. The corresponding histograms for the standard deviations of the individual measurements, $\sigma_{WaMoS}$ and $\sigma_{ADCP}$ (WaMoS® II: red, ADCP: blue) show the same behavior: WaMoS exhibits most variation in the interval 0–0.02 m/s, decaying exponentially at higher values. Due to the natural variability of currents, $\sigma > 0$ must be

expected, especially in areas with strong currents. Therefore, $\sigma$ does not solely reflect precision of the measurements. Again, both components (*UE, UN*) show the same behavior. The distribution of the ADCP data has its peak at the 0.02–0.04 m/s bin and decays from there exponentially to higher values. For both sensors, the individual standard deviations $\sigma_{WaMoS}$ and $\sigma_{ADCP}$ are below the estimated common value, $\sigma_\Delta$, obtained from the comparison: $\sigma_{WaMoS}$, $\sigma_{ADCP}$ < $\sigma_S$. The square root of the sum of the squares of both standard deviations is also below the combined value: $\sqrt{\sigma^2_{WaMoS} + \sigma^2_{ADCP}}$ < $\sigma_\Delta$. The slightly lower values of $\sigma_{WaMoS}$ compared to $\sigma_{ADCP}$ are assumed to result from the different spatial coverages. Since the ADCP delivers measurements that are locally more confined than those obtained by WaMoS® II, they can be assumed to be subject to more variability than the WaMoS® II data representing areal means over several square kilometers. In summary, most of the observed deviations between WaMoS and ADCP can be explained by the different observation volumes (vertical and horizontal extents) and the natural velocity variability contained therein. The results of the comparison between WaMoS and ADCP for the different current components are summarized in Table 1.

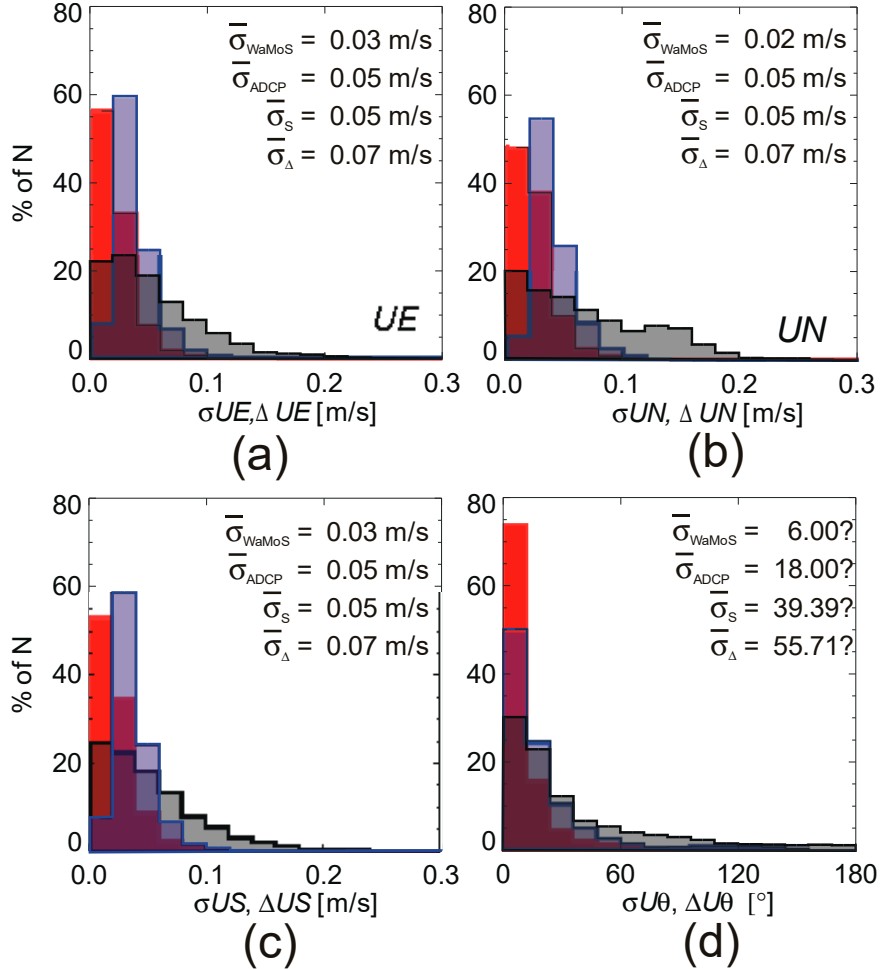

**Figure 10.** Histogram of the standard deviation of the mean WaMoS® II (red) and ADCP (blue) current component—(**a**) *UE*, (**b**) *UN*, (**c**) *US*, and (**d**) *Uθ*—and the absolute difference between the WaMoS and ADCP measurements (black). The bin width of the histograms is 0.02 m/s for the current components and 12° for the current direction.

**Table 1.** Results of the comparison between WaMoS® II surface and ADCP subsurface current measurements.

| Parameter | Symbol | UE | UN | US | Uθ |
|---|---|---|---|---|---|
| Number of data sets | N | | 7272 | | |
| Correlation coefficient (Equation (1)) | $r$ | 0.94 | 0.97 | 0.96 | 0.87 |
| Bias (Equation (2)) | $\overline{\Delta}$ | −0.02 m/s | −0.06 m/s | −0.0004 m/s | −6.88° |
| Total standard deviation of the difference (Equation (3)) | $\sigma_{\Delta}$ | 0.07 m/s | 0.07 m/s | 0.07 m/s | 55.71° |
| Individual standard deviation (Equation (4)) | $\sigma_S$ | 0.05 m/s | 0.05 m/s | 0.05 m/s | 39.39° |
| Standard deviation of the temporal mean ADCP measurements (Equation (5)) | $\sigma_{ADCP}$ | 0.05 m/s | 0.05 m/s | 0.05 m/s | 18.00° |
| Standard deviation of the temporal WaMoS®II measurements (Equation (5)) | $\sigma_{WaMoS}$ | 0.03 m/s | 0.02 m/s | 0.03 m/s | 6.00° |

## 7. Summary and Conclusions

The key motivation of this analysis was to evaluate the usability of the MR-based *sigma* S6 WaMoS® II system with respect to the reliability, precision and eventually accuracy of the surface current measurements. As previous evaluations of WaMoS® II measurements were based on direct comparisons with ADCP measurements, the resulting accuracies often lead to misinterpretation. This is because the data sets used are results of different measurement principles, and also because of temporal and spatial misalignment of the data sets [24]. The fact that vertical and horizontal current shears, which lead to deviation of WaMoS® II surface and ADCP subsurface current measurements, do not automatically reflect an error in one of the measurements was occasionally mentioned in earlier work [15,25].

To reduce the effect of natural current variability on the data set comparison, temporal means over an averaging interval of 20 min were used. The averaging further allows one to estimate the standard deviation of the observed current. The mean current values of WaMoS® II and ADCP were then directly compared with standard statistical tools, such as correlation coefficient ($r$), bias ($|\overline{\Delta}|$), and standard deviation ($\sigma_{\Delta}$, $\sigma_s$). The results of $r > 0.9$, $|\overline{\Delta}| < 0.06$ m/s and $\sigma_s = 0.05$ m/s reveal an excellent agreement between the two data sets and hence the validity of the measurements, especially when taking into account that these values may include deviations unrelated to errors or inaccuracies in the measurement devices but to vertical and horizontal inhomogeneities. Only data sets acquired in the equatorial region, where a large vertical current shear associated with the Equatorial Undercurrent exists, were excluded from this comparison. The standard deviation of the individual mean current value ($\sigma_{WaMoS}$, $\sigma_{ADCP}$) was assumed to reflect the precision of the individual measurements, even when this parameter is strongly linked to the natural variability of the currents. Here the results $\sigma_{WaMoS} = 0.02$ m/s for both current components reflect the high stability of the measurement during different current regimes and wave conditions. This value represents a combination of the natural variability of the flow and potential measurement errors. Due to the spatial character of the WaMoS® II measurement $\sigma_{WaMoS}$ is lower than the equivalent value obtained for the ADCP, $\sigma_{ADCP} = 0.04$ m/s. The fact that both $\sigma_{WaMoS}$ and $\sigma_{ADCP}$ do not exceed the combined single standard deviation $\sigma_s$ confirms the consistency of the validation. Assuming that the sensor-related error of the ADCP is equivalent to the theoretical error $\sigma_{ADCP(theoretical)} = 0.0113$ m/s the error related to the natural current variations $\sigma_{fluctuation}(ADCP) = \sigma_{ADCP} - \sigma_{ADCP(theoretical)} = 0.03$ m/s. The fact that $\sigma_{WaMoS} < \sigma_{fluctuation}(ADCP)$ is related to the larger integration domain of the WaMoS and that $\sigma_{ADCP(theoretical)}$ is an ideal value, which in reality is likely larger.

To ensure consistent data set precision, and that all WaMoS® II data sets satisfy this high validity, an internal quality control flag is set for each individual measurement. During *Polarstern* cruise PS113

different environmental conditions with high and low wind speeds and wave heights as well a different precipitation conditions were met, proving the proper performance of the WaMoS® II *rtQC*.

For operational use of X-Band radar wave and current observations, stringent quality control is advisable. Interferences in the radar images or in the absence of wind no reliable and accurate observation of waves and current are possible and can lead to inaccurate measurements. Therefore, the quality of the measurement needs to be indicated to the user.

For our comparison of WaMoS® II measurements with ADCP reference data, we unfortunately could not make use of ADCP currents above 17 m. This points to a general shortcoming of vessel-mounted ADCPs; namely, that no current measurements are taken from the water column above the keel depth of the ship plus some blanking distance. This shortcoming hampers scientific progress in the understanding of processes which govern the coupling of atmosphere and ocean. Having shown in our study that quality-controlled WaMoS® II measurements compare well with ADCP data in regions which are not obviously subject to strong near-surface vertical shear such as, for instance, the equatorial region with its undercurrent, suggests making use of WaMoS® everywhere when the quality control indicates valid measurements, and combine those with ADCP data in order to obtain a full vertical current profile reaching up to the surface.

**Author Contributions:** Conceptualization, K.G.H.; methodology, K.G.H. and W.-J.v.A.; software, K.G.H. and W.-J.v.A.; validation, K.G.H. and W.-J.v.A.; formal analysis, K.G.H. and W.-J.v.A.; investigation, K.G.H., S.E.N., W.-J.v.A., and V.H.S.; resources, K.G.H., S.E.N., W.-J.v.A., and V.H.S.; data curation, K.G.H. and W.-J.v.A.; writing—original draft preparation, K.G.H.; writing—review and editing, K.G.H., S.E.N., W.-J.v.A., and V.H.S.; visualization, K.G.H.; supervision, K.G.H.; project administration, S.E.N. and V.H.S.; funding acquisition, K.G.H., S.E.N., and V.H.S.

**Funding:** This research received no external funding.

**Acknowledgments:** The authors would like to thank the German Weather Service *DWD* for providing the wind information. Furthermore, we would like to thank Ralf Krocker (AWI) and Thomas Liebe and Johannes Rogenhagen (*F. Laeisz*) for supporting the WaMoS® II installation on board *Polarstern*. ADCP Software OSSI, version 1.9 provided by T. Fischer, GEOMAR, Kiel. We are moreover grateful to the comments provided by four reviewers, which helped to improve the manuscript.

**Conflicts of Interest:** The authors declare no conflict of interest.

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
