# Peer review of "On the Reliability of Surface Current Measurements by X-Band Marine Radar"

_remotesensing, doi:10.3390/rs11091030_

Round 1

Reviewer 1 Report

This paper represents a worthwhile and useful contribution to the field of radar oceanography, introducing some interesting techniques for comparing area-based and in-situ surface current measurements. I find no reason why this paper should not be accepted for publication and find no fault with its contents, although a fine comb-through for any spelling errors may be warranted.

Author Response

Dear Reviewer,

Thank you very much for your fast and positive review. We are grateful that you spent your time on it and recommended our manuscript for publication. Because you had no specific comments, we do not provide a point-by-point response here. However, during revision of the manuscript, we followed your suggestion and put careful attention to the English language, hoping to have removed any spelling errors now.

Kind regards

Reviewer 2 Report

The manuscript “On the reliability of surface current measurements by X-band marine radar”, Hessner et al. analyze a large wealth of data collected under varying conditions. It is intended to asses te accuracy and reliability of current measurements as estimated by a specific remote sensing system based on X-band data, WaMoS II. 

The analysis presented is based on several criteria to discard data collected under conditions where signal to noise ratios we’re not suited to perform the measurements. Next, data deemed appropriate were contrasted with an in-situ sensor showing a reasonable agreement. The authors argue that this is proof of the good performance of the system.

While in scope the work as presented could be suited to Remote Sensing, I struggle to see the scientific value of it, considering that it is very sensor-specific. Alas, I think it would be best suited as a technical report or manual for WaMoS than a research article. At most, I would consider it a technical communication.

For instance, the Quality Control section is very terse in the detail to be of any use by a third party. While it is possible to understand the rationale behind the checks, details on the processing of the data are largely omitted so replicability of this is not possible. It is understandable that this quality control could be presented as a distinctive feature of WaMoS, but if that is the main goal I would consider it would be appropriate to publish it elsewhere.

Analysis of the current itself follows the basic comparison strategies, although some details are missing or mixed without any particular reason. For instance, in Eq. 2, the bias is estimated using signed values, which could be subject to cancelling leading to low values if error estimates follow Gaussian statistics. Perhaps in these early assessments absolute values should be used instead. On the other hand, absolute values are introduced when graphing the bias, in which case it would be desirable to use signed values to assess how the errors distribute about the mean. 

Alternatively, the use of a standard metric as the root-mean-square-error is notoriously absent. 

There are aspects that are  not clearly explained nor analyzed how they affect the data. For instance,  temporal averages are used, but no sensitivity analysis is carried out regarding how the length of the averaging window affects the estimates. Analysis of direction estimates is also absent. On the other hand, sometimes speculative comments are presented without a substantial basis, such as suggesting that turbulence is the cause of non-zero errors. 

The article contains several typos and some grammar errors (for instance, data is treated as singular). I am not detailing them here because I would recommend proof-reading.

The bottom line is that I am struggling to find this manuscript publishable as a research article. I think it would require a change of focus or reorganizing it, adding more in depth analysis to be suitable for publication.

Author Response

Revision Report: (reviewer comments in plain style, responses in italics, text changed/added in manuscript boldface)

Dear Reviewer,

Thank you very much for fast your review. We are grateful for your detailed comments and suggestions, which helped us to improve our manuscript.

.

Stimulated by your final conclusion. we highlight the scientific value of our manuscript and make this more prominent in the revised version.

We think that the combination of the different aspects of WaMoS measurements discussed in our manuscript namely:

- Quality control

- Measurement accuracy

- Measurement precision

is novel and interesting to the readers and not covered by other publications. In general, the measurement accuracy of a sensor is estimated by direct comparison with reference data only. These results give only an indication of the real accuracy of a measurement, as deviations of the different data sets do not automatically reflect measurement errors or inaccuracies. Here we compare the measurement agreement of two data sets with the consistency of the individual sets. We were able to show that a stringent quality control is required when using remote sensing technology like WaMoS which is based on standard marine X-Band radar for scientific applications. The presented data analysis demonstrate that WaMoS data which pass the internal real time quality control (rtQC) are accurate (with respect to ADCP measurements) and precise (low temporal standard deviation) and hence are reliable data for different kinds of application including scientific research.

Moreover, we are convinced that our new results are of wider interest for the oceanographic community and should therefore be published in a scientific journal and not be hidden in a technical report or alike. To make this clear we have added to the end of the manuscript the following sentences:

For our comparison of WaMoS® II measurements with ADCP reference data we unfortunately could not make use of ADCP currents above 17 m. This points to a general shortcoming of vessel-mounted ADCPs, namely that no current measurements are taken  from the water column above the keel depth of the ship plus some blanking distance. This shortcoming hampers scientific progress in the understanding of processes which govern the coupling of atmosphere and ocean. Having shown in our study that quality-controlled WaMoS® II measurements compare well with ADCP data in regions which are not obviously subject to strong near-surface vertical shear as, for instance, the equatorial region with its undercurrent, suggests to make use of WaMoS® everywhere when the quality control indicates valid measurements, and combine those with ADCP data in order to obtain a full vertical current profile reaching up to the surface.”

With respect to specific comments (here summarized in bullet points), here are our detail response:

·         For instance, the Quality Control section is very terse in the detail to be of any use by a third party. While it is possible to understand the rationale behind the checks, details on the processing of the data are largely omitted so replicability of this is not possible. It is understandable that this quality control could be presented as a distinctive feature of WaMoS, but if that is the main goal I would consider it would be appropriate to publish it elsewhere.

In order not to be too technical, we just briefly describe the general approach and the key steps of the applied Quality Control (rtQC). We added some more information related to the rtQC including a flow chart, which should give a better overview. In our opinion, quality control is a key feature in the processing chain of any kind of data, especially when using remote sensing data. Its acquisition can be subject of various sources of interferences not common in in-situ measurements. Therefore, a stringent applied QC is essential for the use and acceptance of new remote sensing technologies like WaMoS. We think that this topic is rarely described and needs to be stressed and addressed.

Furthermore, data from new sensors needs to be validated before they can be used for scientific interpretations of e.g. the ocean. This needs to happen in a scientifically rigorous way. Our understanding from the editor was that this is welcome in Remote Sensing (please see also our response above).

·         Analysis of the current itself follows the basic comparison strategies, although some details are missing or mixed without any particular reason. For instance, in Eq. 2, the bias is estimated using signed values, which could be subject to cancelling leading to low values if error estimates follow Gaussian statistics. Perhaps in these early assessments absolute values should be used instead. On the other hand, absolute values are introduced when graphing the bias, in which case it would be desirable to use signed values to assess how the errors distribute about the mean. 

For estimating the accuracy of the WaMoS measurement, we used on purpose the signed difference between WaMoS and ADCP. To clarify this, we added two new figures (Figure 8b,d ,9b,d) showing signed histograms of the probability distribution of the difference (DUE, DUN, DUS, DUq) to the manuscript and histograms of current speed and direction). These reflect the near Gaussian distribution of the differences.

·         Alternatively, the use of a standard metric as the root-mean-square-error is notoriously absent. 

We avoid using the terminus root-mean-square-error as we think that this parameter does not reflect a real error of the presented measurements. For the presented data set, we think that this parameter specifies the general deviation of the data sets, which are not automatically related to errors, rather to natural variabilities in the current field. Therefore, we use only correlation, bias and standard deviation to describe the agreement of the data sets.

·         There are aspects that are  not clearly explained nor analyzed how they affect the data. For instance,  temporal averages are used, but no sensitivity analysis is carried out regarding how the length of the averaging window affects the estimates. Analysis of direction estimates is also absent. On the other hand, sometimes speculative comments are presented without a substantial basis, such as suggesting that turbulence is the cause of non-zero errors. 

It is our aim to resolve changes in the ocean that take place on spatial scales of several kilometers and temporal scales of hours to days. These are e.g. fronts, inertial oscillations, mesoscale and submesoscale flows. Conversely, we do not want to explicitly resolve the short term (on the order of seconds to a few minutes) velocity fluctuations that are due to e.g. turbulence and surface waves. Therefore, 20 minutes is an averaging interval that falls in between what we want to resolve and what we want to average over. Our statistics would not change significantly, if e.g. we would use 15 minutes or 25 minutes, but they would for shorter or longer periods as we would then not be able to select only the specific processes of interest.

We have added an explicit presentation of direction (and also absolute speed) differences.

We did not explain the temporal averaging in detail. In Lines 89-92 we state the relation between average and fluctuating part of a property. The length of the averaging interval needs to be chosen carefully, so that natural fluctuations due to short-term current variabilities (turbulences) are averaged out while long-term variations of the current are still captured.

·         The article contains several typos and some grammar errors (for instance, data is treated as singular). I am not detailing them here because I would recommend proof-reading.

We have proofread the manuscript and hope to have removed all errors.

·         The bottom line is that I am struggling to find this manuscript publishable as a research article. I think it would require a change of focus or reorganizing it, adding more in depth analysis to be suitable for publication.

We added a broader focus to the paper, and added some closing sentences to the end of the paper pointing out its wider scientific value as explained above. But we also quote from the journal’s aims and scope (https://www.mdpi.com/journal/remotesensing/remotesensing_flyer.pdf): “Remote Sensing (ISSN 2072-4292) publishes regular research papers, reviews, letters and communications covering all aspects of remote sensing science, from sensor design, validation/calibration, to its application in geosciences, environmental sciences, ecology and civil engineering. Our aim is to publish novel/improved methods/approaches and/or algorithms of remote sensing to benefit the community, open to everyone in need of them.” (emphasis added). Hence we believe that the present manuscript---even though it has a technical focus---falls within what is supposed to be published in Remote Sensing.

Kind regards

Reviewer 3 Report

The paper is well written and organized. But, I haven't found any new point for quality control. I think I cannot suggest publication for this paper.

Author Response

Revision Report: (reviewer comments in plain style, responses in italics, text changed/added in manuscript boldface)

Dear Reviewer,

Thank you very much for your fast review and the time you spent on this.

Referring to your final comment: “But, I haven't found any new point for quality control. I think I cannot suggest publication for this paper.”, we are highlighting the scientific value of our manuscript. We think that the combination of the different aspects of WaMoS measurements discussed in our manuscript namely:

- Quality control

- Measurement accuracy

- Measurement precision

is novel and interesting to the readers and not covered by other publications.

In our opinion, quality control is a key feature in the processing chain of any kind of data, especially when using remote sensing data. Its acquisition can be subject to various sources of interferences not common in in-situ measurements. Therefore, a stringent applied QC is essential for the use and acceptance of new remote sensing technologies. We think that this topic is rarely described and needs to be stressed and addressed.

Further, the measurement accuracy of a sensor is commonly estimated by direct comparison with reference data from sensors with known accuracy. These results give only an indication of the real measurement accuracy, as deviations of the different data sets do not automatically reflect measurement errors or inaccuracies. Here we compare surface currents (WaMoS) with subsurface (ADCP) currents. The data was collected in various current and sea state regimes with highly variable current fields, on the Southern and Northern Hemisphere. Vertical and horizontal shear are expected and we take account of this.

The temporal consistency is described by the standard deviation of a sliding mean. Assuming small short distance variation in the ocean currents, this gives an indication for the WaMoS II precision.

We are sure the new points in our quality control become more clear with the the following flow chart that we included in the revised version of the manuscript:

Figure 2: Flow chart of the WaMoS II real time quality control (rtQC). The different quality checks (diamonds) are carried out with in the different levels of the WaMoS processing chain. Boxes indicate key processing steps from the radar images in polar coordinates to the resulting wave spectra, peak wave and current parameters.

In order to stress the wider scientific value of our study, we have added the following sentence to our manuscript:

For our comparison of WaMoS® II measurements with ADCP reference data we unfortunately could not make use of ADCP currents above 17 m. This points to a general shortcoming of vessel-mounted ADCPs, namely that no current measurements are taken  from the water column above the keel depth of the ship plus some blanking distance. This shortcoming hampers scientific progress in the understanding of processes which govern the coupling of atmosphere and ocean. Having shown in our study that quality-controlled WaMoS® II measurements compare well with ADCP data in regions which are not obviously subject to strong near-surface vertical shear as, for instance, the equatorial region with its undercurrent, suggests to make use of WaMoS® everywhere when the quality control indicates valid measurements, and combine those with ADCP data in order to obtain a full vertical current profile reaching up to the surface.”

Reviewer 4 Report

This manuscript describes the differences of two systems measuring current speed and direction along a transatlantic cruise of Polarstern. The authors have described the systems as well as the data, the proposed filtering and the comparison sufficiently well. However, improvements can be made in presenting the procedure, e.g. through a flowchart which summarizes the data flow and processing. My detailed comments are in the attached pdf-copy. The manuscript would benefit from proof-reading, just use the reference section as an example of the abundance of typos.

Based on the references list, Remote Sensing may not be the best journal for this manuscript, and remote sensing principles are not really explained herein, it is more a comparison of two sensors, which leads me to suggest that submitting it to MDPI Sensors may be an option. In fact, the manuscript should be published in a journal which the users of  WAMOS II would frequently read. I will leave this up to the editor to decide.

In summary, I think that the manuscript should be published after a mayor revision.

Author Response

Revision Report: (reviewer comments in plain style, responses in italics, text changed/added in manuscript boldface)

Dear Reviewer,

Thank you very much for your fast review and the time you spent on this. We greatly appreciate your detailed comments, which we used to revise our manuscript thoroughly. We are sure our  revised paper benefited a lot from you comments and suggestions.

Besides our direct reponses to your comments, we try to highlight the scientific value of our manuscript.

We think that the combination of the different aspects of WaMoS measurements discussed in our manuscript namely:

- Quality control

- Measurement accuracy

- Measurement precision

is novel and interesting to the readers and not covered by other publications. In general, the measurement accuracy of a sensor is estimated by direct comparison with reference data only. These results give only an indication of the real accuracy of a measurement, as deviations of the different data sets do not automatically reflect measurement errors or inaccuracies. Here we compare the measurement agreement of two data sets with the consistency of the individual sets. We were able to show that a stringent quality control is required when using remote sensing technology like WaMoS which is based on standard marine X-Band radar for scientific applications. The presented data analysis demonstrate that WaMoS data which pass the internal real time quality control (rtQC) are accurate (with respect to ADCP measurements) and precise (low temporal standard deviation) and hence are reliable data for different kinds of application including scientific research.

Moreover, we are convinced that our new results are of wider interest for the oceanographic community and should therefore be published in a scientific journal and not be hidden in a technical report or alike. To make this clear we have added to the end of the manuscript the following sentences:

For our comparison of WaMoS® II measurements with ADCP reference data we unfortunately could not make use of ADCP currents above 17 m. This points to a general shortcoming of vessel-mounted ADCPs, namely that no current measurements are taken  from the water column above the keel depth of the ship plus some blanking distance. This shortcoming hampers scientific progress in the understanding of processes which govern the coupling of atmosphere and ocean. Having shown in our study that quality-controlled WaMoS® II measurements compare well with ADCP data in regions which are not obviously subject to strong near-surface vertical shear as, for instance, the equatorial region with its undercurrent, suggests to make use of WaMoS® everywhere when the quality control indicates valid measurements, and combine those with ADCP data in order to obtain a full vertical current profile reaching up to the surface.”

With respect to specific comments (here marked with bullet points), here are our detail response below:

·         L37: Define weather clutter, what is it, rain, fog etc?

We think a full definition of weather clutter is not needed here, as here we wanted to stress only that other signatures can disturb the sea clutter observations. So we changed the sentence to:

“Also, signatures of rain or snow (weather clutter), or other features in the radar image, which are not related to sea clutter, can disturb MR wave and current observations.”

Later in the text we explain the rain signatures in more detail (see Figure 3 and Figure 4)

·         L136 Georeferencing should be explained a little better, what are the input parameters, how is a fixed position determined/defined? Any references for this method which seems to be implemented in the wamos system?

We add two references describing the concept of georeferencing. Further, we add this.

“The motion compensation in space-time domain is done by georeferencing [15] [19]. Using GPS ship position and heading (gyro), orientation and position for every radar pulse are estimated. When transforming the sea clutter information from polar coordinates to Cartesian image sequences, each point of the resulting analysis area corresponds to a fixed position relative to the earth, independent of how the vessel is moving during the acquisition time.”

·         L170 Is this sufficient to avoid directional dependency? Why would you not determine the wave orientation and then re-project all three images accordingly to get the true wavelength? Maybe I misunderstood, but averaging over only three spectra seems insufficient.

We have good experience with this approach, as the analysis areas can be placed in all directions so it can be assumed that the resulting spatial mean spectrum represents the full area of radar footprint.

However, using different analysis areas has also other advantages compared to analyzing the entire radar image at once. It does not only account for spatial inhomogeneity due to directional imaging mechanisms (described in detail in the paper by Lund et at.), but it also allows to take care of spatially localized disturbances in the radar view field or inhomogeneous wave fields, which otherwise can lead to misleading results. In coastal areas for example, processes like wave refraction, shoaling, wave current interaction lead to spatially varying wave fields (described in Hessner et al…). In these cases, spatial homogeneity over the entire radar footprint cannot be assumed.

Location and size of the analysis areas can be configured with respect to the customers need. This is of practical use as WaMoS is an operational system applied do different kinds of application and operation sites.

Another advantage of using different analysis areas also in homogeneous wave fields, like it is in the presented application, is the possibility to quality check the results of the individual areas before spatial averaging. Temporary unreliable results in a particular area can be rejected while the results of the other areas allow continuous observations.

·         L191 The ADCP this paragraph is important for your study, but there is not enough information on how the STD is derived nor how the matlab tool box works. At the least, you need to provide references which contain more details. If the ADCP processing is done via a black box, the study will become affected by unknown processing of your calibration/comparison data.

Here we explained in more detail the used ADCP data and add some references.

As a reference, data from a vessel mounted acoustic Doppler current profiler (ADCP) type Ocean Surveyor from Teledyne RD instruments [21] were used. Its transducers/receivers, operating at a nominal frequency of 150 kHz, are mounted in the hull of Polarstern, about 11 m below the water line. It was working in long-range mode with a vertical cell size of 4 m, a blanking distance of 4 m, and a maximum range of ~320 m. Heading, pitch, and roll from the ship’s inertial navigation system (GPS and magnetically constrained “gyro”) were used to convert the ADCP velocities to earth coordinates. The accuracy of the ADCP velocities mainly depends on the quality of the position fixes and the ship’s heading data. Further errors stem from a misalignment of the transducer with the ship’s centerline. The ADCP data were processed using the Ocean Surveyor Sputum Interpreter (OSSI) developed by GEOMAR, Helmholtz Centre for Ocean Research, Kiel ( [21]) which corrects for a possible misalignment between the ADCP transducer orientation and the ship’s forward direction.

 In order to avoid interference with vessel-induced currents, the ADCP measurements are averaged over the 20-50 m depth range. For the data comparison, quality controlled ADCP current data with averages over 2 minutes were used. The quality filter is based on the statistical analysis, where data outliers exceeding the range of mean value and standard deviation of surface velocity are neglected. The ideal-theoretical precision  of the ADCP measurements can be estimated from the single ping/bin standard deviation of 0.3 m/s, given by the ADCP manufacturer. Neglecting natural variability and assuming vertical and temporal homogeneity and independence over 20-50 m (7 depth bins) and 2 minutes (100 pings), results in  0.0113 m/s.

Further we want to state  that the OSSI software is widely used by the oceanographic community (e.g. von Appen et al. 2018).

·         L225: This manuscript would benefit from a flow chart outlining the two data streams, filtering, quality checks and comparisons made. It is very difficult to follow only the text. A flow chart would be very helpful.

Thanks to the reviewer’s helpful suggestion, we added the following chart and caption.

Figure 2: Flow chart of the WaMoS II real time quality control (rtQC). The different quality checks (diamonds) are carried out with in the different levels of the WaMoS processing chain. Boxes indicate key processing steps from the radar images in polar coordinates to the resulting wave spectra, peak wave and current parameters.

·         Figure 2: where is the rain signatures, name the color or area of rain affected regions. I guess the cyan colored region?

We mark the rain signatures in the images and in the text we clarify that rain signatures are visible as patches of high backscatter intensity.

·         L262: Are you using partial images for the analysis, or is a full image either used or rejected? If you use partial data, e.g. one of the three sections, it would affect your averaging?

The WaMoS data analysis including rtQC is carried out individually for each analysis area hence not on the full images. We are confident this becomes more obvious with the flow chart (Figure 2). For the spatial averaging all wave spectra with insufficient quality (IQ<10) are rejected, hence do not contribute to the resulting mean. By allowing only data which pass the data bases quality control, inaccuracies due to partly insufficient or disturbing backscatter and ensure optimal performance are minimized. Nevertheless, the rejecting of information from particular areas lowers the significance of the results. Therefore, in the data basis check (last check) the number of used wave spectra are tracked.

·         L306 why is the current speed so high, if it is simply random scattering from a smooth surface, the current speed should not be amplified. where in your algorithm does the processing allow this to become large with no evidence? is it rain which creates high speeds randomly?

When no sea clutter is visible in the radar image, like in the absence of sufficient wind (e.g. May 18th) the radar images still contain signal associated with thermal noise. This noise is not related to the actual sea state and current and hence does not contain any information on it. When the WaMoS algorithm then tries to find a current solution the results can be anything and in particular also high current speeds, which are not related to the real surface current. This is also the reason why the results scatter randomly across the full range. The same holds for strong rain signatures, which cover the wave information in the sea clutter. In operation, unreliable data which do not pass the rtQC are rejected and not displayed. Only for this publication, we kept these results to show how false results look like and to demonstrate the capability of the rtQC.

·         Figure7: why not zoom in to -2 to 2 range of both x and y axes.

We changed the range of the plot to [-2m:2m]

·         not sure how and IQ limit of 10 was justified?

We evaluated the limit IQ = 10 empirically. Meaning in our evaluation, it turned out that also data with 0 <IQ <10 yield to accurate and precise hence reliable results. For example if the sea clutter check of the rtQC identifies a missing image in the analysis sequence or partially disturbed area by rain or interferences, it does not automatically lead to corrupt results. Furthermore, it turned out the WaMoS system is more robust to partly missing data. Before that evaluation is was not clear, how sensitive the WaMoS current retrieval is to specific disturbances in the sea clutter. For practical applications the IQ limit for the data basis check, can be configured so that the final data bases check in rtQC can be adjusted to the used radar and station set up.

·         To clarify this we add this paragraph before the flow chart.

For the presented data set it turned out that the sea clutter checks do not indicate unreliable data on their own. This means that partly missing or disturbed sea clutter information alone does not automatically lead to identification as unreliable data. Only in combination with the subsequent  quality checks, can the results be regarded as unreliable. Even when the results of the sea clutter checks do not explicitly indicate insufficient data, it reveals the potential cause of measurement failures. This will be discussed in more detail with respect to rain signatures later on. For practical use an IQ valid limit of 10 is set.

Finally, we want to state that we added a broader focus to the paper, and added some closing sentences to the end of the paper pointing out its wider scientific value as explained above. But we also quote from the journal’s aims and scope (https://www.mdpi.com/journal/remotesensing/remotesensing_flyer.pdf): “Remote Sensing (ISSN 2072-4292) publishes regular research papers, reviews, letters and communications covering all aspects of remote sensing science, from sensor design, validation/calibration, to its application in geosciences, environmental sciences, ecology and civil engineering. Our aim is to publish novel/improved methods/approaches and/or algorithms of remote sensing to benefit the community, open to everyone in need of them.” (emphasis added).

Hence we believe that the present manuscript---even though it has a technical focus---falls within what is supposed to be published in Remote Sensing.

Kind regards

Round 2

Reviewer 2 Report

This is the second review of the  manuscript “On the reliability of surface current measurements by X-band marine radar”, Hessner et al. . The authors have thoroughly attended most of my previous comments, and in general, the manuscript is more robust.  I consider it is suitable for publication once these very minor corrections are atended, which I do not need to review again.

1.L238 and thereabouts: I think it would be very useful to have an example of the kind of codes can be obtained. For example, how a data seta that failed two or three of the system checks is identified.

2. L 253: The limit of 10 is in binary or decimal notation?

L437: between *and* Wamos and ADCP. Remove extra “and”.

Author Response

Dear Reviewer

Thank you very much for your fast second review and that you spent again time on our manuscript. Following your suggestions, we implemented some minor changes to the manuscript, which are clarified below,

Kind regards

1.      L238 and thereabouts: I think it would be very useful to have an example of the kind of codes can be obtained. For example, how a data seta that failed two or three of the system checks is identified.

Accordingly, we have added the following sentences:

P6 Li 251-256 “For example, if the sea clutter test yields that parts of the analysis area contain rain signatures (see Figure 1), this leads to IQ = 4. If in the SNR check the data does not pass the significance test so that IQ = 20, this is added to give IQ = 24. Further, if the wave system check failed as the resulting spectrum is too noisy to identify significant individual wave systems, IQ = 100 is added. The resulting accumulated then is IQ = 124, indicating unreliable measurements due to insufficient radar data because of rain.”

2.      L 253: The limit of 10 is in binary or decimal notation?

We changed the sentence to:

P7 L 263: “For practical use a decimal IQ valid limit of 10 is set.”

3.      L437: between *and* Wamos and ADCP. Remove extra “and”.

corrected

Reviewer 3 Report

The authors have made a significant improvement of the manuscript. Some points should be clarified or revised before the publication.

P1, Li 31: Give full name of "MR".

P3, Li 101: It should be a regular "Y". 

P3, Li 110: Remove ", P3".

P4, Li 129: Please explain why the MR is insensitive to vessel motions such as pitch, roll and heave. Is it a property of all MR or only the WaMos? Some necessary references should be added.

P10, Li 313: "IQ = [1,2] " seems not a correct expression. 

P15, Fig. 10, the signs of degree in subfigure (d) are not shown correctly.

And, could you give a brief explanation about whether it is necessary to check the other conditions when IQ > 10, since finally all the data when IQ>10 are  discarded.

Author Response

Dear Reviewer,

Thank you very much for your fast second review and that you spent again time on our manuscript. Following your suggestions we implemented some minor changes to the manuscript, which are clarified below.

Kind regards

1.      P1, Li 31: Give full name of "MR".

MR stands for Marine Radars, so we add (MR) in first line of the introduction for clarification.

P1 Li29:   “Marine radars (MR) are designed …”

2.      P3, Li 101: It should be a regular "Y". 

corrected-

3.      P3, Li 110: Remove ", P3".

corrected-

4.      P4, Li 129: Please explain why the MR is insensitive to vessel motions such as pitch, roll and heave. Is it a property of all MR or only the WaMos? Some necessary references should be added.

This is a general property of all MR, as they are built to scan the sea surface independent from the ship pitching and rolling. This is independent from sigma s6 WaMoS.

To clarify this we slightly modified the related sentence and add a reference [19].

P4 LI 129-133: “The large vertical beam width of MRs, which range normally between 20 to 25°, depending of the used radar, ensures the ability to scan the sea surface even when the ship is pitching and rolling [19]. Hence, we assume that vessel motions like pitch, roll and heave have no critical influence on the WaMoS measurements.

[19] A. Noris, A. D. Wall, A. G. Bole and W. O. Dineley, Radar and ARPA Manual: Radar and Target Tracking for Professional Mariners, Yachtsmen and Users of Marine Radar, 2 ed., Elsevier Science, 2005, p. 544.

5.      P10, Li 313: "IQ = [1,2] " seems not a correct expression. 

marked with IQ = [1 or2],

6.      P15, Fig. 10, the signs of degree in subfigure (d) are not shown correctly.

We do not understand this comment. Shown in Fig. 10 are histograms of standard deviations of means, which are always positive. We suspect that something might have went wrong during download of our manuscript by the reviewer. At least, we don’t have any other explanation for this comment.

7.      And, could you give a brief explanation about whether it is necessary to check the other conditions when IQ > 10, since finally all the data when IQ>10 are  discarded.

During our study, it turned out that the sea clutter tests leading to 0 < IQ < 10 do not explicitly identify unreliable measurement results. These tests just indicate that the sea clutter information is somehow affected. Further tests are necessary to assess if the results are reliable or not.

In the manuscript we answer this question – also in response to a comment of reviewer 2 – by adding the following sentences.

P6 Li 251-256 “For example, if the sea clutter test yields that parts of the analysis area contain rain signatures (see Figure 1), this leads to IQ = 4. If in the SNR check the data does not pass the significance test so that IQ = 20, this is added to give IQ = 24. Further, if the wave system check failed as the resulting spectrum is too noisy to identify significant individual wave systems, IQ = 100 is added. The resulting accumulated then is IQ = 124, indicating unreliable measurements due to insufficient radar data because of rain.”

Reviewer 4 Report

Thank you for improving the clarity of the presentation, improving the figures and adding additional details on the technical aspects. I recommend publication of the manuscript.

Author Response

Dear Reviewer,

Thank you very much for your fast second review and that you spent again time on our manuscript. Because you had no specific comments, we do not provide a point-by-point response here.

Thank again for helping to improve our manuskipt.

Kind regards
